# Practitioner's perspectives on access to Rabies Post-Exposure Prophylaxis in Tanzania: A mixed-methods and theoretically-informed study to inform policy and practice

Kennedy Lushasi[1]*, Eleanor M. Rees[2], Camille Barker[2], Nai Rui Chng[2], Ally Halfan[1], Christian Tetteh Duamor[1], Elaine Ferguson[2], Felix Lankester[3,4], Husna Hoffu[1], Joel Changalucha[1], Lwitiko Sikana[1], Mikolaj Kundegorski[2,5], Katie Hampson[2]

**1** Environmental Health and Ecological Sciences, Ifakara Health Institute, Dar es Salaam, Tanzania, **2** School of Biodiversity, One Health & Veterinary Medicine, College of Medical, Veterinary & Life Sciences, University of Glasgow, Glasgow, United Kingdom, **3** Paul G. Allen School for Global Animal Health, Washington State University, Pullman, Washington, United States of America, **4** Global Health Tanzania, Arusha, Tanzania, **5** Fjelltopp Ltd, Mildenhall, United Kingdom

* klushasi@ihi.or.tz

## Abstract

Rabies remains a preventable yet fatal disease that continues to claim lives in low- and middle-income countries due to barriers in accessing post-exposure prophylaxis (PEP). With Gavi, the Vaccine Alliance, now supporting PEP provision in rabies-endemic countries, understanding access barriers is critical. We used mixed-methods to synthesise quantitative data on bite patients (>10,000) captured through an Integrated Bite Case Management platform across four regions of Tanzania, with qualitative data from hotline calls and peer-support chat established among health and veterinary workers from 2018 to 2024. We applied an expanded healthcare access framework comprising six dimensions: availability, accessibility, affordability, accommodation, acceptability, and appropriateness to guide our analyses and assess barriers and facilitators to PEP delivery and uptake in these regions. We identified interconnected barriers to PEP access including recurrent vaccine stockouts (up to 9.2% regionally); long travel distances that were greater for rural patients (on average: 34 vs 13 km for urban patients) who represented the majority at risk; and unaffordable direct and indirect costs leading bite victims to either not start or abandon PEP, incur risky delays or default to traditional remedies. Incorrect bite patient management, including misuse of biologicals sold at unregulated private pharmacies led to preventable deaths, while insufficient vaccination supplies and inadequate service hours, increased risks and costs to patients. Despite these gaps, peer-support mechanisms improved patient management and real-time problem-solving. PEP delivery in Tanzania is undermined by systemic barriers across all access dimensions. Addressing these requires decentralizing vaccine delivery to prioritized accessible facilities, ensuring reliable stock and supplies for intradermal administration, reducing patient

**Data availability statement:** Quantitative data and code to reproduce the results, including the figures and tables are available via a GitHub repository (https://github.com/LushasiK/IBCM-PEP-TZ) archived on Zenodo: https://zenodo.org/records/18508654.

**Funding:** The work was supported by the Wellcome Trust [207569/Z/17/Z and 224520/Z/21/Z to KH], the UK Medical Research Council [MR/Z504919/1 to KH] and the Department of Health and Human Services of the National Institutes of Health [R01AI141712 to FL]. The funders had no role in study design, data collection and analysis, decision to publish, or preparation of the manuscript. No authors received salary support from any of these funders. The content is solely the responsibility of the authors and does not necessarily represent the official views of the National Institutes of Health.

**Competing interests:** The authors have declared that no competing interests exist.

costs, strengthening health worker capacity through training and support, and regulating private providers. Lessons from Tanzania highlight the need for context-sensitive, equity-focused strategies to maximize the impact of Gavi's investment and accelerate progress toward Zero by 30.

## Introduction

Rabies is a neglected zoonotic disease that continues to pose a major public health challenge in low- and middle-income countries (LMICs) [1]. Despite the availability of effective tools for rabies control and prevention, the disease still claims nearly 60,000 human lives globally each year [2]. The majority of these deaths occur among children living in rural communities in Asia (60%) and Africa (36%) [2]. Responding to this longstanding inequity, the Tripartite organizations (World Health Organisations (WHO), World Organisation for Animal Health (WOAH), and the Food and Agricultural Organisations (FAO)) set the global strategic goal "Zero by 30" to end dog-mediated human rabies deaths by 2030. The strategy to achieve this goal is based on increasing awareness of and access to post-exposure prophylaxis (PEP), and scaling up mass dog vaccination to eliminate rabies at its source [3].

In Tanzania, approximately 900 people die from rabies each year [4]. Although fatal once symptoms develop, the onset of rabies can be prevented by prompt administration of PEP [5]. PEP comprises thorough wound washing followed by a series of vaccinations started promptly after an exposure, and, for severe wounds, Rabies Immunoglobulins (RIG) infiltrated into the wound [5]. However, challenges arise for health systems in providing PEP, and for exposed persons in obtaining them. Improved understanding of barriers to accessing PEP and how they can be overcome should be used to guide rabies prevention. Access to healthcare however, as implied by PEP access, is a complex issue. There are numerous conceptual frameworks for understanding access, all with their own strengths and limitations. Levesque *et al* (2013) defines access as "the opportunity to identify healthcare needs, to seek healthcare services, to reach, to obtain or use health care services, and to actually have the need for services fulfilled" [6]. This expanded conceptualisation of access builds upon earlier formulations, where access is described as "the degree of "fit" between the patients and the system" by Penchansky & Thomas (1981) in terms of five dimensions: availability, accessibility, accommodation, affordability, and acceptability [7]. Healthcare access is a broad and complex concept and its definition, dimensions, and influencing factors continue to be debated [8].

Improved healthcare for rabies prevention includes access to PEP as part of a One Health approach that involves cross sectoral collaborations to deliver mass dog vaccination, and targeted education. Surveillance for rabies also requires a One Health approach and for this WHO recommends Integrated Bite Case management (IBCM), a system that links health and veterinary professionals in assessing the rabies risk among bite victims and investigating rabies suspect animals [9]. The recent commitment by Gavi, the Vaccine Alliance, to invest in human vaccines for PEP presents an opportunity to catalyse progress towards the "Zero by 30" goal

[1,10,11]. Gavi's investment is projected to avert nearly 500,000 rabies-related deaths within a 15-year timeframe [4]. Animal investigations as part of IBCM can identify other high-risk bite victims who would not otherwise seek PEP and increase awareness of rabies among communities [9,12]. For these reasons, Gavi also recommends that eligible countries adopt IBCM to improve rabies detection, and increase PEP adherence [13]. Examining healthcare access for rabies prevention is timely in the context of Gavi's investment and the "Zero by 30" global strategy.

Recognizing this public health burden and leveraging Gavi's efforts to increase PEP access in eligible countries, Tanzania applied for Gavi-supported human rabies vaccines. IBCM has been undertaken in four regions of Tanzania since 2018 [9]. Benefiting from this IBCM platform, we draw on health workers' experiences from across these regions, highlighting barriers and facilitators that impact bite victims' access to PEP. We synthesize key lessons to inform efforts to improve PEP access in Tanzania and in other countries that are Gavi-eligible.

## Materials and methods

### Ethics statement

The study was approved by the Medical Research Coordinating Committee of the National Institute for Medical Research of Tanzania (NIMR/HQ/R.8a/vol.IX/2788), the Ministry of Regional Administration and Local Government (AB.81/288/01), the Institutional Ethical Review Board of Ifakara Health Institute (IHI/IRB/No: 22–2014) and the Medical, Veterinary and Life Sciences ethics committee at the University of Glasgow (200240271).

### Study design

We adopted a mixed methods approach using data from the IBCM platform between 2018 and 2024. Records of bite patients attending health facilities enrolled into the platform were explored quantitatively, and communications between practitioners carrying out IBCM were interrogated through a qualitative lens. We adopted Penchansky and Thomas's healthcare access framework due to its more accessible framing, which addresses the limitations of our data (i.e., the lack of qualitative data directly from patients) [7].

### Study area

The study covered four regions of Tanzania (Lindi, Mtwara, Mara and Morogoro, excluding Gairo and Mvomero districts), that comprise 31 district authorities where IBCM has been implemented (Fig 1) [9]. The human population in these regions was reported as 7,718,148 from the 2022 National Census [14]. The average human-to-dog ratio (HDR) is estimated at 24:1, corresponding to a dog population of around 349,500 [15]. However, the HDR varies from just 7.2:1 in Mara to 58:1 in Lindi [15], translating to very different dog population sizes (<21,000 dogs in Lindi versus almost 330,000 in Mara). Economic activities in these regions also vary, with Mara being predominantly agro-pastoralist, Morogoro having a mix of agro-pastoralist societies and peasant farming (small-scale farmers cultivating their land or working land owned by others for subsistence), while Lindi and Mtwara are dominated by peasant farming and fishing [16].

### Integrated Bite Case Management platform

IBCM was introduced in 2018 across the four regions, enrolling district hospitals and health centers that provide PEP [9]. From 2020 the platform was extended to cover facilities identified as new PEP providers or that frequently referred bite patients (including private facilities and newly established district hospitals). Focal persons within enrolled facilities were selected to attend joint training sessions with Livestock Field Officers, covering risk assessments to determine whether patients were bitten by high-risk (potentially rabid) or low-risk (healthy) animals, PEP delivery (Fig 1), investigations of biting animals and data entry [9]. Health workers and Livestock Field officers were requested to electronically submit records

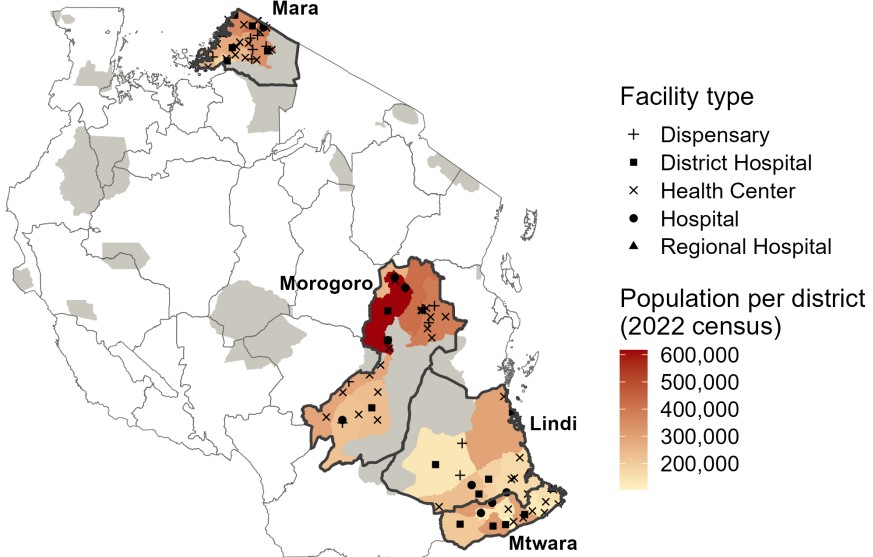

**Fig 1. Regions in Tanzania implementing Integrated Bite Case Management.** Health facilities enrolled into IBCM are shown in black, with the shape indicating the facility type. The red shading indicates the human population density by district, with wildlife-protected areas (where human activities are prohibited) in gray. Data sources: Ward-level population data were downloaded from the Tanzania National Bureau of Statistics GIS portal (https://www.nbs.go.tz/statistics/topic/gis); protected areas were obtained from ArcGIS Online (https://www.arcgis.com/home/item.html?id=780c707de03842a-98a365c30ceef1a74); health facility locations were compiled by the authors and are available via Zenodo (https://zenodo.org/records/18508654).

of risk assessments and animal investigations, respectively, from a bespoke mobile application to an IBCM database hosted at Ifakara Health Institute.

A peer support group was created during the establishment of IBCM to connect health and veterinary workers involved in IBCM. Researchers managed a group chat to facilitate peer-to-peer knowledge sharing, reinforce PEP protocols, and support real-time communication on rabies-related issues. In addition, a free hotline was established by researchers for IBCM practitioners and members of the public to seek guidance on patient management, report rabies incidents, troubleshoot challenges, and access information on rabies. Hotline calls were logged, recording the date-time stamp, sender's cadre, sector and workstation, the nature of their question and the response or support provided (Fig 2).

## Data analysis

We extracted records of patient risk assessments and animal investigations from the IBCM database from June 2018, when IBCM was established, to December 2024. Using these data we quantified the number of patients seeking and completing PEP, delays to PEP administration, distances travelled for PEP (between the centroids of the wards of health facilities and patients' homes), and reported stockouts faced by patients presenting to facilities. We calculated the proportion of PEP delivered via route of administration. For these analyses, we also examined regional variations and assessed temporal trends. These analyses were conducted using R (version 4.4.1) [18], with distances between centroids calculated using the "geosphere" package [19].

Consent was sought from IBCM practitioners (health and veterinary workers) to use their anonymized data on experiences with PEP access from the peer support group chat and hotline log. Messages from the group chat and hotline log were extracted from June 2018 to December 2024, including the date, the sender's professional position, sector, and work location. Identifying characteristics were removed and text translated from Swahili into English.

**PLOS** **Global Public Health**

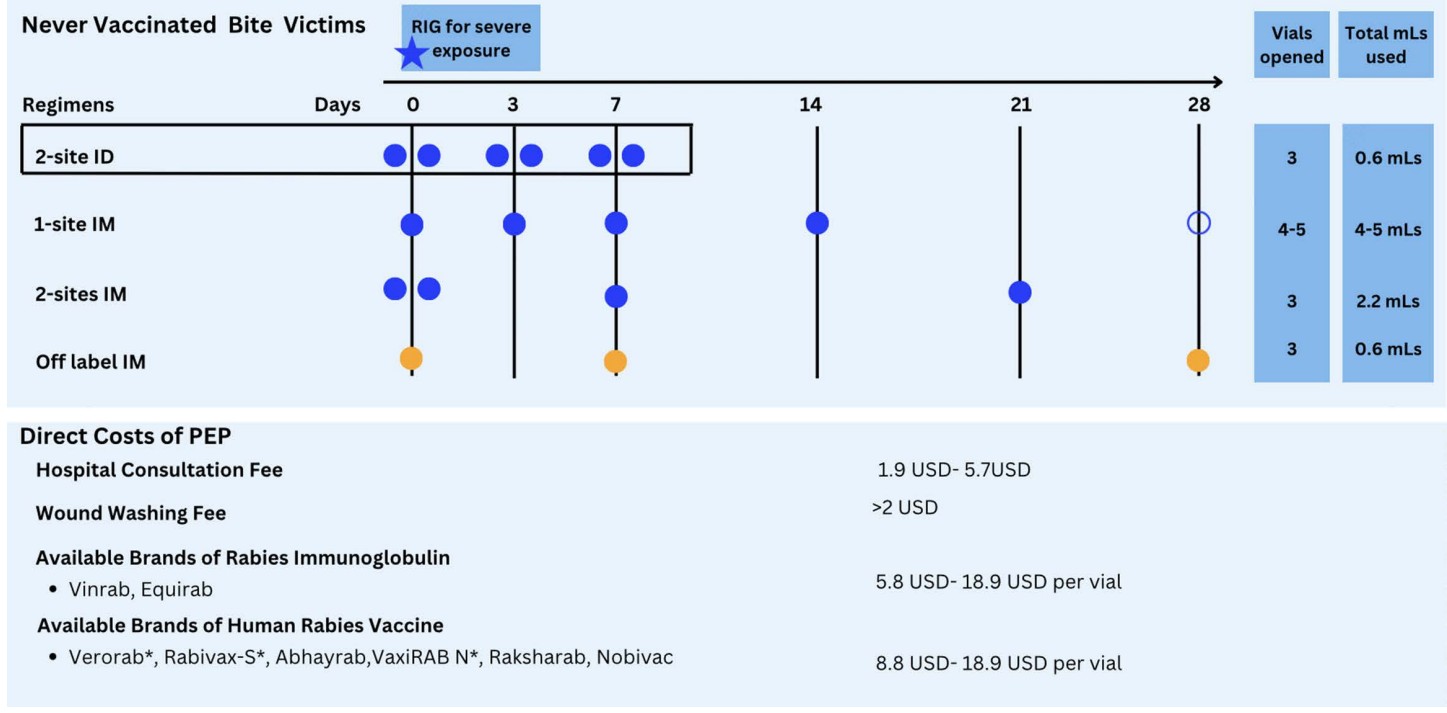

**Fig 2. PEP regimens used in Tanzania and their associated costs.** Each circle on the timeline indicates an injection. ID regimens use 0.1mLs of vaccine at 2 injection sites, while IM regimens' use an entire vial for each injection. WHO recommends adoption of the one-week ID regimen indicated in the box, as well as the two IM regimens coloured in blue [5]. In Tanzania, a 5th dose of the two-week IM regimen (open circle) is also used in some facilities, as well as an off-label IM regimen, shown in orange, that is neither recommended by WHO nor included in the Tanzania standard treatment guidelines. Costs of PEP (minimum to maximum reported) are shown in USD (exchange rate 1USD-2648Tsh [17]), although these vary by facility and required medical attention. WHO pre-qualified vaccines are indicated with an *.

Using NVivo V.14.0, we analyzed messages from the peer support group chat and the hotline log to explore practitioner-centred perspectives related to barriers and facilitators to PEP access. Initial thematic analysis was conducted by applying an inductive approach to categorise these data. As we developed themes, a code-book was developed to define the present themes and adjusted through an interactive process to explore new aspects of the dataset. This process was continued until we reached thematic saturation.

Methodological triangulation [20] was applied to systematically compare the findings between the qualitative and quantitative data using a matrix to compare findings from both data types. Data was grouped in this matrix by relevance to the analytical constructs (Table 1). Discussions among co-authors were held to review findings, and areas of disagreement were resolved by reviewing the data. This approach allowed for understanding of topics that may have been misinterpreted by an individual researcher and to explore topics which may have been overlooked. Synthesis of the data highlighted factors affecting access to PEP which go beyond Penchansky and Thomas' original framework [7], with discussion leading to the defining of a 6th construct, 'Appropriateness', as described in Table 1.

## Results

The IBCM database comprised records of 10,370 bite patient presentations to enrolled health facilities, including details on the administration of PEP to 7,997 patients. We extracted records of 1,070 hotline calls and 2,179 messages from the peer support group chat. We organized our findings below according to the constructs of our theoretical framework.

**Table 1. Constructs used for conceptual framework examining access to care in relation to PEP access in Tanzania. Constructs from Penchansky and Thomas's framework [7] are listed with definitions rewritten to reflect access to PEP. Associated questions are included to contextualize these definitions as applied to PEP access in Tanzania. A new construct 'Appropriateness' was added to capture complexities related to PEP access in Tanzania not well covered by the original framework.**

| Construct | Definitions | Questions |
|---|---|---|
| 1. Availability | The relationship between the health system's management of health services and the patient's ability to receive necessary care. | To what extent can the health system reliably supply PEP to meet demand? |
| 2. Accessibility | Geographic accessibility - how easily the patient can physically reach the provider's location | How difficult is it for bite victims to reach a facility that provides PEP and what means do they have to support this health-seeking? |
| 3. Affordability | The relationship between the direct costs of healthcare and indirect costs of care seeking, and the patient's ability to pay. | What are the costs associated with obtaining PEP and how do these impact bite victim's care seeking? |
| 4. Accomodation | The extent to which the provider's operation is organized in ways that meet the constraints and preferences of the patient. | Which operational characteristics of health facilities facilitate/ impede a patient from seeking and receiving adequate PEP? |
| 5. Acceptability | The extent to which the patient is comfortable with the more immutable characteristics (e.g., age, gender, type of facility) of the provider, and vice versa. | How comfortable are bite victims with the kind of care they received, including PEP? |
| 6. Appropriateness (new) | The extent to which health providers understand the required health service procedures and are able to provide appropriate care to patients. | To what degree are health workers able to interpret and perform procedures for administering appropriate PEP? |

## Availability

Among 7,997 patients recommended for PEP, 5% faced stockouts. Regional variation in patients encountering stockouts ranged from 1.7% in Lindi to 9.2% in Mara, with Mara recording the largest absolute number of shortages, impacting 189 bite patients (Table 2). 124 human rabies deaths were reported during the study period. Reports from the hotline log and peer support group conversations indicated that some deaths resulted from stockouts:

> "The dog was killed but she never started the vaccine because there was none at the health facility, they later brought her back to the facility with symptoms of rabies." [Hotline log 79, HW 29, Region 2]

Stock management issues were frequently reported, with the most acute problems occurring in rural areas, according to qualitative data:

> "Because most of these incidents are in rural areas, but mostly vaccines are available in urban areas; a person decides to convince himself that there is no problem. The result is loss of life." [Peer Support Chat 1185, HW 67, Region 4]

**Table 2. Challenges to high-risk bite patients initiating and completing PEP. Here we define PEP completion as having received at least 3 doses, irrespective of the regimen used.**

| Region | PEP shortage | PEP completion (3rd/ 1st dose) | ID route of administration (ID/ total) | Mean delay in days to first dose (95% CI) | ≥2 days delay (%) |
|---|---|---|---|---|---|
| Lindi | 1.66% (20/1204) | 30.07% (356/1184) | 51.18% (606/1184) | 2.32 (1.78-2.85) | 21.26% |
| Mara | 9.17% (189/2062) | 40.79% (764/1873) | 65.31% (1220/1868) | 1.56 (1.19-1.93) | 15.43% |
| Morogoro | 3.87% (164/4234) | 36.5% (1484/4066) | 63.4% (2578/4066) | 2.53 (2.16-2.9) | 23.47% |
| Mtwara | 5.49% (28/510) | 30.66% (145/473) | 60.25% (285/473) | 3.38 (1.89-4.86) | 21.84% |

However, even in district hospitals, rabies vaccine shortages were common, in contrast to other routinely provided vaccines.

> "Where are vaccines available? Because here he has been injected with tetanus [toxoid] only, even at the district hospital vaccines are not available." [Peer Support Chat 557, HW 112, Region 4]

Shortages reportedly resulted in bite victims seeking care from private pharmacies, which led to further issues (see Appropriateness):

> "The challenges we are facing is that health centers in the periphery have no anti rabies vaccine; a large percentage [of patients] end up in pharmacies." [Peer Support Chat 1158, LFO 9, Region 2]

Additionally, qualitative data suggests that switches between ID vs IM vaccination were partly due to lack of insulin syringes with fine (high-gauge) needles needed for ID administration:

> "The problem is the availability of the syringes for injecting ID. Maybe the big facilities that give insulin might have them." [Peer Support Chat 509, HW 14, Region 4]

The extent to which the health system reliably supplies PEP (Availability) also depends on demand, which is influenced by patients' knowledge and awareness and their ability to reach and pay for services (see Accessibility and Affordability). Delays to PEP were more common for bites from known dogs (32.4% of bites from known dogs; n = 634) compared to unknown dogs (20.6% of bites from unknown; n = 5,197) and reports from the peer support chat suggest that bites from dogs known to the victim may influence their care seeking decisions:

> A child was bitten by a dog suspected of having rabies.. he was not taken to the hospital to get vaccinated because they knew the dog, and thought it was only a small dog so that was harmless…the child showed symptoms of rabies. But still the parents hid… By yesterday afternoon he had died. [Peer Support Chat 1762, LFO 66, Region 3]

High-risk patients may be aware of the risks of rabies, and seek care, but costs remain an obstacle to completing PEP (see Affordability), forcing some to turn to traditional remedies.

> A rabid dog bit 9 people, among them, 2 died of rabies. One … was given 2 vaccinations..but the third week did not arrive and he died. He put tobacco leaves on the wound and other natural medicines… There is the issue of education being very low in our communities, but also the availability of vaccinations and their prices are a challenge. [Peer Support Chat 1095, HW 5, Region 2]

However in other examples, low-risk bite patients, who know the vaccination history of the biting animal still seek PEP, because of the reassurance PEP provides.

> "I have been bitten by a dog whose history is known to have been vaccinated but I want to be vaccinated for my safety, am I allowed?" [Hotline log 152, Community Member 5, Region 3].

### Accessibility

The majority of facilities providing rabies vaccine (96%) are located in urban or semi-urban areas, yet most bite patients (69.2%) come from rural areas. Of the 46 health facilities located in rural areas enrolled in IBCM, only 43% (20/46) were still providing PEP by December 2024, due to the high costs of procuring these vaccines, with the remainder instead

referring patients to other facilities. To access PEP, patients from rural areas traveled considerably farther than patients from urban settings, traveling an average of 34.37 km compared with 12.82 km respectively. On average, patients traveled 30.69 km to reach a facility, varying substantially by region, from 17 km in Mara to 52 km in Lindi, reflecting the degree to which PEP was decentralized to local facilities (Fig 3).

> "In remote areas, the journey from there to here is challenging. If these services were distributed to every health facility, it would be beneficial, and we could save many lives." [Peer Support Chat 1171, HW 14, Region 2]

Moreover, stockouts forced patients to travel even greater distances or endure (risky) delays while waiting for resupply:

> "The patient gets a referral to the Regional Hospital, they get money for anti-rabies but they can't manage the cost to the city, so by cooperating with the DMO's office we procure and keep the anti-rabies at the village Dispensary." [Peer Support Chat 934, LFO 27, Region 4]

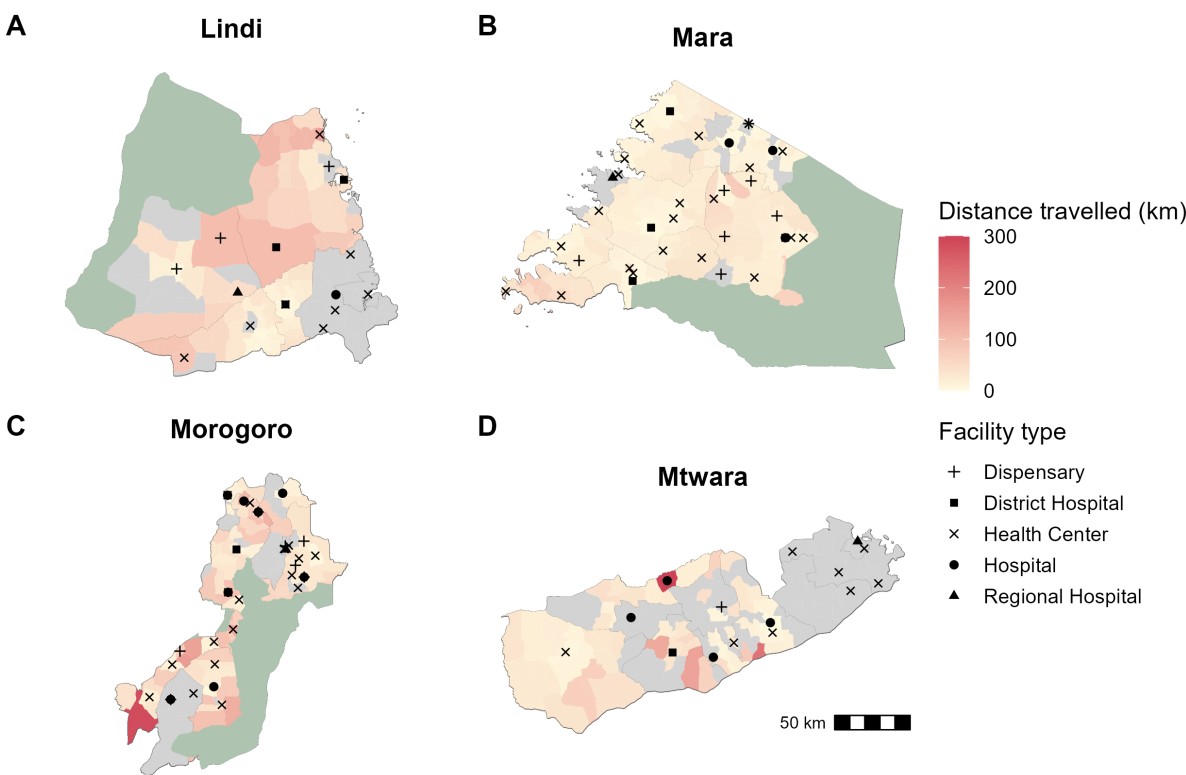

**Fig 3. Distances travelled by patients seeking PEP.** Ward polygons are shaded according to average distance travelled by bite victims from that ward. The black symbols indicate the location and type of health facilities known to stock PEP undertaking IBCM. Grey shaded wards had no recorded bite patients. The green shading indicates protected areas. Data sources: Ward-level population data were downloaded from the Tanzania National Bureau of Statistics GIS portal (https://www.nbs.go.tz/statistics/topic/gis); protected areas were obtained from ArcGIS Online (https://www.arcgis.com/home/item.html?id=780c707de03842a98a365c30ceef1a74); health facility locations were compiled by the authors and are available via Zenodo (https://zenodo.org/records/18508654).

## Affordability

Qualitative data showed that despite some patients being aware of the risks, they struggled to pay for either transport and/ or vaccines leading to delays as they solicited funds.

> "Another person who has not been able to get the vaccination fees... He's trying to sell his cashew farm but has not found a customer. He tells you that 'I really want to get those vaccines but I don't have the money!'" [Peer Support Chat 1093, HW 5, Region 2]

Non-initiation or failure to complete PEP was common and often attributed to lack of money, with on average only 36.2% patients finishing three or more vaccine doses (Fig 4a and Table 2), and no improvement over the study period. Cost barriers sometimes had fatal consequences, as illustrated by this incident where a rabid dog bit multiple individuals:

> "I gave advice and emphasized that they should follow up on [PEP] vaccinations, but they neglected the advice, giving economic reasons. There were those who received one dose and others did not receive any injection at all! One month later, two died and the doctors confirmed that it was rabies." [Peer Support Chat 1253, LFO 43, Region 3]

When faced with these combined costs, some bite victims refused care despite health providers outlining the risks:

> "An issue is the costs; many rabies victims struggle to complete their vaccines. Our patients come from low-income backgrounds, so despite educating them, affording the anti-rabies treatment can be challenging for them." [Peer Support Chat 1762, HW 322, Region 1]

Whereas others resorted to traditional remedies after learning about vaccination costs despite being informed of risks:

> "I went to a village to follow up on a jackal case and I found an old man sticking leaves on his wound after knowing the cost of the vaccination and it [Rapid Diagnostic Test] was positive." [Peer Support Chat 1092, LFO 15, Region 3]

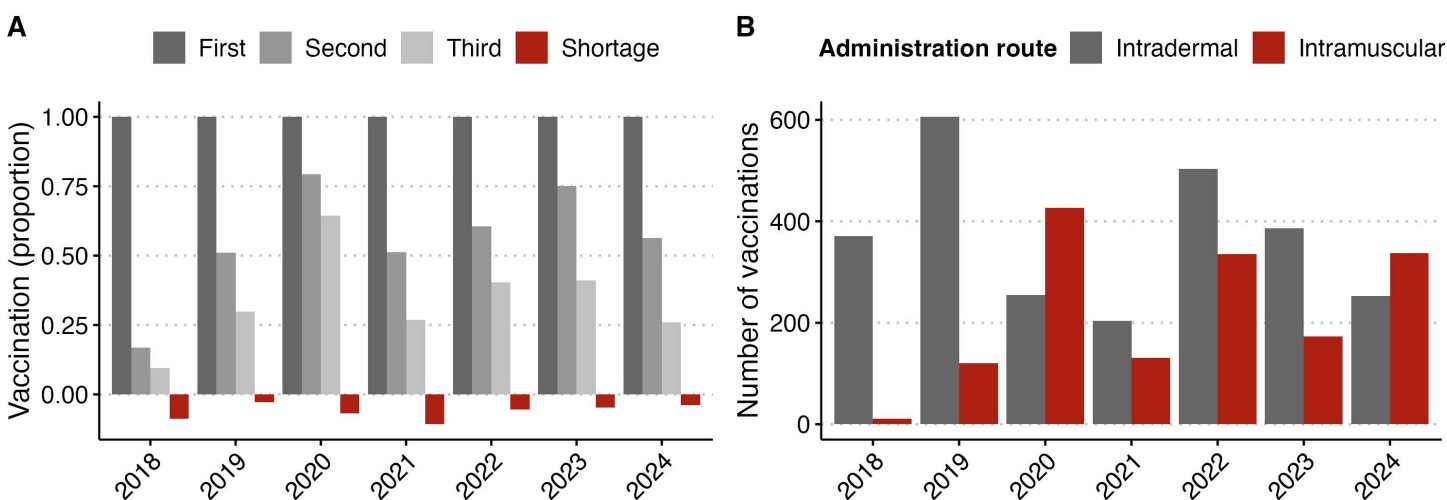

**Fig 4. Animal bite patients receiving PEP by year. A)** Proportion of bite victims receiving first, second or third doses of PEP, with patients facing stockouts indicated in red. **B)** Number of PEP doses delivered by year, via intradermal or intramuscular route.

## Accomodation

Qualitative data suggested that the way in which health clinics are operated did not meet the needs of bite victims. Clinics that do not operate during the weekend can either delay patients or even cause them to give up on PEP:

"She visited the clinic the day she was bitten but the [clinic that provides vaccines] was closed because it was a Saturday. They did not come back until it was too late." [Hotline log 740, HW 102, Region 2]

While unanticipated closures of clinics also affected patients' completion of PEP:

"A patient was delayed in his second dose because the Reproductive and Child Health Immunisation department (RCH) was closed when he arrived and he did not come back. He came to [the hospital] today with rabies symptoms and died." [Peer Support Chat 805, HW 18, Region 3]

## Acceptability

We have no empirical data on 'Acceptability' as a dimension of access because this theme is dependent on patient perspectives which our data source does not contain.

## Appropriateness

Quantitative data showed that while 51.7% of PEP doses were delivered via the ID route, this varied across regions (Table 2) and by year, with ID use dropping from 95.4% in 2018 to 42.5% in 2020 (Fig 4b), sometimes as insulin needles were not available (see Availability). PEP completion rates were similar irrespective of vaccination route (39.0% for ID and 32.5% for IM).

Qualitative data revealed knowledge gaps among health workers in following PEP protocols which affected the quality of patient care. For example, some health workers reported difficulties with recalling procedures, and relied on peer support for guidance:

"Guys, please help me with that chat on how to provide vaccination through ID, I have forgotten, please remind me" [Peer Support Chat 475, HW 49, Region 3]

Others sought guidance on vaccine scheduling and clarification on the use of RIG.

"I have forgotten the schedule for vaccinating after a dog bite. When should the patient follow up? When do we use RIG?"[Hotline Log 89, HW 20, Region 4]

Knowledge gaps of health workers were compounded by practices of private pharmacies, who were observed to sell RIG as an alternative to rabies vaccines:

"... the doses available in the [private] pharmacy are labelled Immunoglobulin, which I have never treated anyone with, what should I do?"[Hotline Log 749, HW 12, Region 4].

The emergence of private pharmacies as stop-gap when patients faced stockouts at health facilities was problematic. Follow up visits by researchers observed examples at pharmacies of improper vaccine storage, incorrect vaccine administration, and even fraudulent practices with other products missold as rabies vaccines.

"There are also pharmacies where untrusted employees have made these vaccines as their source of income, even if the patient does not have the criteria to be injected, after taking the history they inject TT [Tetanus Toxoid] VACCINES two doses and tell them that he has completed all his doses, something that is more DANGEROUS to health" [Peer Support Chat 1068, HW 126, Region 2].

One health worker described patients receiving unsafe products at pharmacies due to inadequate regulation and oversight:

"Private pharmacies must be monitored because many people are deceived into thinking they are receiving help but are instead harmed." [Peer Support Chat 1185, HW 144, Region 4].

Purchase of RIG had not previously been seen in Tanzania. But, visits by a private sales representative to hospitals, health facilities and pharmacies in multiple regions of Tanzania from late 2022 led to local procurement of RIG. Since then misuse of RIG as a substitute for rabies vaccine began to be reported, with examples of some health workers (incorrectly) believing these products were interchangeable:

"Several times I quarreled with my head of department when he insisted we use the Equirab because he thinks it works the same as anti-rabies vaccines. But I kept on educating him." [Peer Support Chat 944, HW 23, Region 3]

Researchers and health officials used the peer support chat to explain the difference between these products, and share guidance to address failings that had led to fatalities:

"...He received three scheduled doses [photo of RIG included]. Now the patient is starting to show signs of rabies...why did this happen?" [Peer Support Chat 1094, HW 92, Region 2]

## Discussion

### Key findings

We identified a range of interconnected barriers across the health system and within populations that hinder access to PEP for rabies prevention in Tanzania. Using a modified access to healthcare framework, we found that PEP availability was limited by stockouts, with access still challenging even when patients had knowledge and awareness, and for bite victims in rural areas PEP was particularly hard to reach. Affordability was the most evident obstacle, preventing patients from both initiating and completing PEP. Health services did not accommodate the emergency needs of bite victims and health practices were not always appropriate. Most concerning was the misuse of biologicals, including confusion between vaccines and immunoglobulins that precipitated fatalities. Although bite victims typically followed health provider advice, we also found that awareness of rabies risks was sometimes lacking, or that cost barriers led victims to default to traditional healers or delay or abandon PEP. We do not have any direct patient data to show that Acceptability was a factor in PEP access.

### Broader context

Inequities in PEP access mirror challenges widely documented across rural Africa and Asia [21,22]. Persistent vaccine shortages were identified as a critical supply-side barrier to PEP access, consistent with findings from other settings [21,23,24]. Centralization of PEP services increases the burden on rural communities, who paradoxically face higher risks of exposure from the presence of more free-roaming and unvaccinated dogs [25–27]. Rural health facilities were

most vulnerable to stockouts due to supply chain inefficiencies and lack of budget for PEP procurement [28,29], leading victims to travel to more distant facilities, forgo PEP, or resort to unregulated pharmacies. Routine vaccination clinics typically operate only during standard working hours (8 am -3 pm), providing just a narrow 8-hour window to initiate PEP, leaving those seeking care after working hours or on weekends at risk of delays. These systemic barriers contributed to the unregulated private pharmacies emerging as alternative sources for PEP, raising concerns about vaccine quality, and correct PEP provision.

Financial constraints represent a universal barrier to PEP access across rabies-endemic countries. Prohibitive costs prevent timely PEP initiation and completion [25,30,31]. Even when victims recognise the importance of PEP, costs can be prohibitive [2,32,33]. Combined costs of vaccines, travel, and lost work time prevent many from completing PEP [26,30,34,35]. Victims who lack immediate access to funds may need time to raise funds, leading to further delays. Even when PEP is provided free of charge, the indirect out-of-pocket expenses remain substantial [32,33]. The costs of repeat travel and fees deter completion of multi-dose regimens, consistent with reasoning for poor PEP compliance reported in other regions [34,36–38].

Evidence from other African countries shows the urgent need to decentralize health services and strengthen rural healthcare infrastructure [28,39,40]. Such measures would reduce travel burdens, encourage timely PEP access, and address risks posed by unregulated pharmacies. The critical time for bite victims to seek care is immediately after exposure [41]. However, delays are commonly reported in rabies-endemic countries, influenced by cultural beliefs, misinformation about rabies transmission, perceived risk of exposure as well as cost barriers. Our findings are consistent, highlighting delays in seeking care, particularly following bites from known animals, and uncertainty about vaccine necessity [42–44]. One reason may be that people may not consider their own domestic animals as sources of disease [41].

Our findings on ID vaccination suggest progress towards adopting WHO-recommended regimens in Tanzania, which could improve both cost-effectiveness and vaccine availability [29,45]. One obstacle to health facilities using ID vaccination was shortages of appropriate syringes [46]. Additionally, peer-support channels revealed knowledge gaps among health workers regarding ID administration, mirroring challenges documented in other settings [47,48]. The most concerning findings were reports of tetanus toxoid or RIG being mistakenly administered instead of rabies vaccines, highlighting the need for regular training and real-time support for healthcare workers in addressing their uncertainties.

RIG is a recommended component of post-exposure prophylaxis for severe (category III) exposures [5], when used correctly in conjunction with rabies vaccine. But our findings show that its recent introduction in Tanzania occurred in a context of very limited provider familiarity and weak regulation of private pharmacies. As a result, RIG was sometimes misconstrued or mistaken as a substitute for rabies vaccine rather than an adjunct, leading to inappropriate clinical decisions and, in some cases, fatal outcomes. The availability of RIG through private suppliers appeared to widen existing inequities, exacerbating already high out-of-pocket costs, while diverting patients away from standardized public-sector PEP pathways. Without safeguards to prevent substitution or misuse, RIG risks undermining confidence in rabies prevention efforts and obscuring the life-saving role of timely vaccination.

## Strengths and limitations

The primary strength of our study lies in the comprehensive mixed-methods approach to examining access to PEP within a sound theoretical framework. Combining extensive quantitative IBCM data on barriers to PEP access with qualitative insights from practitioners brought understanding about real-world impacts and the adaptive responses of both health workers and patients. The IBCM peer support network and rabies hotline proved valuable as both data collection tools and intervention mechanisms, improving health worker engagement and PEP practices. The established communication channels demonstrated their value in improving knowledge sharing and problem-solving among health workers. Such

approaches could also prove valuable for monitoring and evaluating Gavi's investment in human rabies vaccines to ensure highlighted gaps are addressed, and PEP practices are improved.

However, our study had several limitations. First, in terms of theoretical framework, our data collection methods were not designed to capture perspectives of bite patients when seeking care, which has resulted in a lack of data on 'Acceptability' as a dimension of access preventing us from evaluating 'Acceptability' of rabies PEP provision from the perspective of patients. Second, only health facilities designated to provide human rabies vaccines were included within IBCM, and participation from some of these facilities was inconsistent. Facilities that stopped stocking rabies vaccines or experienced supply shortages often discontinued recording details of bite victims that they referred elsewhere for PEP. Perspectives from these facilities were therefore only captured intermittently if they resumed service provision. Furthermore, during on-site visits, we discovered instances where some health workers had conducted patient risk assessments and provided PEP without recording data. Most critically, the IBCM platform only holds data on bite victims who visited health facilities, excluding those who did not use the formal public health sector, instead using traditional practices or private pharmacies, introducing biases and potentially underestimating barriers to PEP. In addition, IBCM data does not collect patient-level data on the direct or indirect costs of PEP, preventing a quantitative assessment of how financial burden influenced access to and completion of PEP. Furthermore, the historical absence of rabies RIG from routine PEP delivery limited our ability to assess RIG-related barriers across multiple access dimensions beyond provider knowledge and quality of care. Lastly, travel distances between bite incident locations and PEP-providing facilities were calculated as the euclidean distances without accounting for travel routes and conditions. Poor road networks, difficult terrain and infrastructure limitations prevalent in rural areas likely pose additional challenges to accessing timely healthcare.

## Conclusion and recommendations

Our findings lead to recommendations to maximize the impact of Gavi's investment to support rabies PEP provision in eligible countries [11]. We show that there is an urgent need to address structural supply-side barriers to PEP access spanning vaccine availability, equity in geographic access and affordability, and training and support to strengthen healthcare worker capabilities. The confluence of identified barriers present critical opportunities to transform PEP access through targeted health system strengthening [11,49,50], as outlined in Table 3, but require reflexive learning and adaptation to ensure that such systemic changes deliver their intended impacts. A structured approach to making human rabies vaccines free at point-of-care and establishing processes for robust vaccine procurement and supply chain management would dramatically improve access and ensure uninterrupted vaccine availability [22]. Support should extend to provision of appropriate syringes for ID administration and inclusion of ID protocols on vaccine packaging to facilitate broader adoption. Real-time inventory tracking could minimize stockouts and optimize distribution [29]. Rural communities would most benefit from PEP service provision at decentralized health facilities, and could leverage the infrastructure and routine outreach activities strengthened through the Expanded Programme on Immunisation (in Tanzania, known as Immunization and Vaccine Development). To guide implementation, a defined population threshold (e.g., > 50,000 people) and/or distance (e.g., > 15 km) could be used to designate facilities to stock and administer rabies vaccines. Selection of these facilities should consider accessibility (including availability beyond standard working hours), trained personnel, and necessary infrastructure for safe vaccine storage and delivery (including cold-chain and water supply). Community engagement should aim to raise awareness on the importance of prompt PEP, even for bites from known animals and should be tailored to address harmful beliefs. Public health messaging should focus on the life-saving benefits of prompt PEP and health worker training should focus on risk assessment, PEP protocols and patient management. These coordinated interventions, implemented systematically, could greatly improve PEP access and reduce rabies deaths.

**Table 3. Challenges to PEP access and recommendations for improvement.** We show the challenges we identified according to our modified expanded healthcare access framework. We note the multiplicative and multifaceted nature of barriers (leading to a confluence of health system failures and fatal consequences or extreme difficulties for patients) requiring a wholesale approach to improving access, with the requirement for the proposed recommendations to be delivered together.

| Construct | Description of challenge | Recommendation |
|---|---|---|
| Affordability | The high costs of vaccines is an obstacle to health facilities stocking them, particularly facilities in more rural settings, with more limited budgets. Moreover, the high costs of vaccines within facilities, as well as travel (and accommodation) costs, are major barriers for patients, increasing the chance that patients do not start PEP, or increasing risks caused by delays as patients source funds or their non-completion of PEP. | Ring-fence sufficient national budget for rabies vaccines as an essential medicine.<br>Provide rabies vaccines free-at-point of care.<br>Support free PEP provision at designated decentralized facilities to reduce indirect patient costs. |
| Availability | Frequent stockouts increase the difficulty for patients of accessing PEP, extending travel distances and costs, reducing the numbers of patients who initiate and subsequently complete regimens, and leading patients to resort to alternatives sources of care, including unregulated pharmacies and traditional healers. While ID regimens are more resilient to vaccine stockouts as vials can be used for more patients, supplies of insulin syringes also need to be sufficient. | Improve supply chains for PEP (e.g., integrate management and distribution with strengthened IVD/EPI supply chains for routine childhood vaccines)<br>Implement real-time monitoring of vaccine use and alerts for re-stocking [29].<br>Adopt ID regimens and ensure insulin syringe supplies. |
| Accessibility | Facilities stocking PEP are typically located in urban centres only (96%), whereas most rabies exposure occur in rural settings (69%) far from facilities stocking PEP. The average patient travel distances to facilities with PEP are very far (varying between 17–52 km by region) contributing to the difficulties in patients accessing PEP. | Select designated decentralized facilities to stock PEP using defined criteria (e.g., population &/ distance thresh-olds), ensuring accessibility with adequate staffing and infrastructure for safe vaccine delivery (including cold-chain capacity). |
| Accommodation | Facilities not operating on weekends is an obstacle to patients in search of PEP. More generally, after hour access was limited to only a few urban centre health facilities. | Identify means of providing emergency PEP, out-of-hours or on weekends (e.g., through a hotline for patients to arrange emergency access and/or via health centres with 24/7 access). |
| Appropriateness | Health workers were not familiar with vaccination regimens, with many not knowing of the difference between RIG and vaccine, compounded by recent aggressive marketing of RIG by private suppliers, lack of government regulation on PEP suppliers, guidance on PEP components and instruction on appropriate PEP delivery. | Provide up-to-date and regular refresher training and job aids for health workers and support for continued good practices (e.g., peer-support chat and emergency hotline).<br>Improve signposting to public health facilities for PEP.<br>Strengthen regulation of private suppliers. |
| Acceptability | NA – data not available from patients' perspective. | NA |

## Acknowledgments

We are grateful to the government of the United Republic of Tanzania for supporting this research, particularly the Ministries of Health, of Local and Regional government, and of Livestock Development and Fisheries, and the One Health Coordination section under the Prime Minister's office for invaluable support and cooperation. We thank the district authorities involved in IBCM implementation and the dedicated district health and veterinary workers, colleagues at Ifakara Health Institute for support.

## Author contributions

**Conceptualization:** Kennedy Lushasi, Eleanor M Rees, Camille Barker, Nai Rui Chng, Katie Hampson.

**Data curation:** Kennedy Lushasi, Eleanor M Rees, Camille Barker, Nai Rui Chng, Ally Halfan, Christian Tetteh Duamor, Husna Hoffu, Joel Changalucha, Lwitiko Sikana, Mikolaj Kundegorski, Katie Hampson.

**Formal analysis:** Kennedy Lushasi, Eleanor M Rees, Camille Barker.

**Funding acquisition:** Felix Lankester, Katie Hampson.

**Investigation:** Kennedy Lushasi, Ally Halfan, Husna Hoffu, Lwitiko Sikana.

**Methodology:** Kennedy Lushasi, Eleanor M Rees, Camille Barker, Nai Rui Chng, Christian Tetteh Duamor.

**Project administration:** Kennedy Lushasi.

**Resources:** Katie Hampson.

**Supervision:** Nai Rui Chng, Katie Hampson.

**Validation:** Kennedy Lushasi, Nai Rui Chng, Felix Lankester, Joel Changalucha, Katie Hampson.

**Visualization:** Kennedy Lushasi, Eleanor M Rees, Camille Barker, Nai Rui Chng, Elaine Ferguson, Katie Hampson.

**Writing – original draft:** Kennedy Lushasi.

**Writing – review & editing:** Kennedy Lushasi, Eleanor M Rees, Camille Barker, Nai Rui Chng, Christian Tetteh Duamor, Elaine Ferguson, Felix Lankester, Husna Hoffu, Joel Changalucha, Lwitiko Sikana, Katie Hampson.

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
