## [Decision Letter · Decision Letter 0]

12 Jan 2026

PGPH-D-25-03424

Access to Rabies Post-Exposure Prophylaxis in Tanzania: A mixed-methods and theoretically informed study to inform policy and practise

Dear Dr. Lushasi,

Thank you for submitting your manuscript to PLOS Global Public Health. After careful consideration, we feel that it has merit but does not fully meet PLOS Global Public Health’s publication criteria as it currently stands. Therefore, we invite you to submit a revised version of the manuscript that addresses the points raised during the review process.

I would like to sincerely apologise for the delay you have incurred with your submission. It has been exceptionally difficult to secure reviewers to evaluate your study. We have now received three completed reviews; the comments are available below. The reviewers have raised significant scientific concerns about the study that need to be addressed in a revision.

Please revise the manuscript to address all the reviewer's comments in a point-by-point response in order to ensure it is meeting the journal's publication criteria. Please note that the revised manuscript will need to undergo further review, we thus cannot at this point anticipate the outcome of the evaluation process.

We look forward to receiving your revised manuscript.

Kind regards,

Miquel Vall-llosera Camps

Staff Editor

Journal Requirements:

i. Please clarify all sources of financial support for your study. List the grants, grant numbers, and organizations that funded your study, including funding received from your institution. Please note that suppliers of material support, including research materials, should be recognized in the Acknowledgements section rather than in the Financial Disclosure.

ii. State the initials, alongside each funding source, of each author to receive each grant. For example: "This work was supported by the National Institutes of Health (####### to AM; ###### to CJ) and the National Science Foundation (###### to AM)."

iii. State what role the funders took in the study. If the funders had no role in your study, please state: “The funders had no role in study design, data collection and analysis, decision to publish, or preparation of the manuscript.”

iv. If any authors received a salary from any of your funders, please state which authors and which funders.

2. We ask that a manuscript source file is provided at Revision. Please upload your manuscript file as a .doc, .docx, .rtf or .tex.

3. Please ensure that the Title in your manuscript file and the Title provided in your online submission form are the same.

4. Thank you for uploading your study's underlying data set. Unfortunately, the repository you have noted in your Data Availability statement does not qualify as an acceptable data repository according to PLOS's standards.

5. Some material included in your submission may be copyrighted. According to PLOS’s copyright policy, authors who use figures or other material (e.g., graphics, clipart, maps) from another author or copyright holder must demonstrate or obtain permission to publish this material under the Creative Commons Attribution 4.0 International (CC BY 4.0) License used by PLOS journals. Please closely review the details of PLOS’s copyright requirements here: PLOS Licenses and Copyright. If you need to request permissions from a copyright holder, you may use PLOS's Copyright Content Permission form.

Potential Copyright Issues:

a. Figures 1 and 3: please (a) provide a direct link to the base layer of the map (i.e., the country or region border shape) and ensure this is also included in the figure legend; and (b) provide a link to the terms of use / license information for the base layer image or shapefile. We cannot publish proprietary or copyrighted maps (e.g. Google Maps, Mapquest) and the terms of use for your map base layer must be compatible with our CC-BY 4.0 license.

Reviewers' comments:

Reviewer's Responses to Questions

**Comments to the Author**

1. Does this manuscript meet PLOS Global Public Health’s publication criteria? Is the manuscript technically sound, and do the data support the conclusions? The manuscript must describe methodologically and ethically rigorous research with conclusions that are appropriately drawn based on the data presented.

Reviewer #1: Yes

Reviewer #2: Yes

Reviewer #3: Yes

2. Has the statistical analysis been performed appropriately and rigorously?

Reviewer #1: I don't know

Reviewer #2: Yes

Reviewer #3: N/A

3. Have the authors made all data underlying the findings in their manuscript fully available (please refer to the Data Availability Statement at the start of the manuscript PDF file)?

Reviewer #1: Yes

Reviewer #2: Yes

Reviewer #3: Yes

4. Is the manuscript presented in an intelligible fashion and written in standard English?

Reviewer #1: Yes

Reviewer #2: Yes

Reviewer #3: Yes

5. Review Comments to the Author

Reviewer #1: the authors have done appreciable work in collecting data on the health care issues wrt to Rabies prevention in Tanzania. I have the following observations.

Some more statistics ie tables etc may be added to the article to justify the results and discussion.

The tables should then be discussed in detail with conclusion pointing out the steps needed to be taken to improve the prevalent chaos in their health care system.

costs per patient could be estimated inclusive of loss of wages etc. this will further improve the impact of the article.

I couldnt open the supplementary GITHUB site for more results as it seemed to be a paid site.

Reviewer #2: REVIEW

of the article "Access to Rabies Post-Exposure Prophylaxis in Tanzania: A Mixed-Methods and Theoretically Informed Study to Inform Policy and Practice"

This article is a comprehensive study addressing a critical global public health issue: access to rabies post-exposure prophylaxis (PEP) in Tanzania. The study adheres strictly to scientific research principles and is highly relevant, clearly structured, and methodologically rigorous. The use of mixed methods (quantitative analysis of large-scale data from the Integrated Bite Case Management (IBCM) platform and qualitative analysis of healthcare worker communications) within a theoretical model of access to healthcare is a key strength of the study, allowing us not only to document the facts but also to understand the mechanisms and context of barriers. This article makes a significant contribution to both the academic understanding of health systems issues in low- and middle-income countries and to intervention planning, particularly in light of the launch of funding for PEP programs by GAVI (the Vaccine Alliance).

The relevance of this study is undeniable. Rabies remains a deadly, yet 100% preventable, disease that claims tens of thousands of lives annually, primarily among the poorest rural populations in Africa and Asia. The announcement of the "Zero by 30" goal and GAVI's decision to support the procurement of rabies vaccines create a historic window of opportunity. At this moment, field-based evidence is crucial, demonstrating that the mere availability of vaccines in a country is insufficient. This study brings this issue to the forefront by systematically identifying a cascade of interconnected barriers.

The scientific novelty lies in the following:

Application and development of the theoretical model. The authors not only utilize Penchansky and Thomas's classic model (five dimensions of access), but also empirically substantiate and add a sixth dimension—"Appropriateness." This crucial conceptual addition highlights provider competence as a separate, critical access factor, as clearly demonstrated by data on the inappropriate use of RIG or tetanus toxoid.

Depth and Richness of Data. The combination of data on over 10,000 patients with an analysis of real-life healthcare worker conversations and concerns (>3,000 messages) provides a unique, comprehensive picture of the situation. This allows us to move beyond dry statistics ("5% experienced interruptions") to understanding the human tragedies and professional dilemmas behind these figures.

Documenting Emerging Threats. The study documents the emergence of a new, dangerous practice—the use of rabies immunoglobulin (RIG, Equirab) as a substitute for vaccination, which has resulted in fatalities. This is a crucial observation for surveillance systems and regulatory agencies.

The study's methodological framework is exemplary for public health research.

Design: Mixed methods are entirely justified for studying the complex phenomenon of access. The quantitative component objectively captures the scale of the problem (distances, completion rates, interruptions), while the qualitative component reveals the causes, context, and subjective experiences.

Theoretical Framework: The use of a structured conceptual framework (expanded healthcare access framework) lends systematicity to the analysis, prevents fragmented findings, and allows for the classification of barriers into clear categories, which is useful for developing targeted interventions.

Triangulation: The authors consciously employ triangulation, systematically comparing findings from quantitative and qualitative data. This enhances the validity and reliability of the results, allowing for identification of areas where the data complement or corroborate each other.

Ethics: The study has all necessary ethical approvals, and the process for obtaining informed consent from participants (healthcare workers) is described.

Despite its undeniable strengths, the study has a number of limitations, some of which the authors honestly and thoroughly describe in the relevant section. This enhances the credibility of the study.

The absence of a patient voice. This is the main methodological limitation, noted by the authors themselves. The study brilliantly analyzes the system from the service provider's perspective, but the "measure" of acceptability—that is, how much patients trust the service, how comfortable they are, and whether the services meet their cultural expectations—remains unexplored. This is an important gap, as cultural beliefs and stigma can be a significant barrier.

Sample bias. The data only cover those who reached official medical facilities included in the KUSU. The most vulnerable groups—those who immediately turned to traditional healers—didn't have the means to even pay for the first visit.

Reviewer #3: The authors employed a mixed-methods approach, integrating quantitative data from the Integrated Bite Case Management platform with qualitative data from hotline calls and peer-support chats among health and veterinary workers, to identify barriers and facilitators affecting bite victims’ access to post-exposure prophylaxis across four regions of Tanzania. This work addresses a critical and timely public health issue, as inadequate knowledge of post-bite treatment and constraints in access to rabies biologics remain major contributors to preventable rabies deaths in dog-endemic regions, including Tanzania. I have just one major concern with the manuscript in its current version. While the authors correctly explain early in the Introduction (line 69 and Fig. 2) that post-exposure prophylaxis (PEP) comprises of immediate wound washing, a series of rabies vaccinations, and administration of rabies immunoglobulin (RIG) for previously unvaccinated patients with severe (Category III) exposures, this framework is not consistently or adequately reflected in the subsequent analysis and interpretation. Specifically, across four of the five dimensions of the expanded healthcare access framework, the discussion focuses exclusively on rabies vaccines, with little to no consideration of rabies immunoglobulin (RIG). For patients with severe (Category III) exposures, RIG is arguably the most critical component of PEP, and its omission substantially weakens the analysis. Issues related to availability, accessibility, and affordability should explicitly address RIG, as barriers to RIG access often differ from—and are more severe than—those associated with vaccines. Without clearly incorporating RIG into these dimensions, the manuscript provides an incomplete assessment of the factors influencing access to life-saving PEP. I understand the challenges surrounding RIG access, particularly as the authors allude to in line 363, where they note that the purchase of RIG had not previously been seen in Tanzania. Given this context, the implications of RIG unavailability should be discussed explicitly and in greater depth in the Discussion section. Doing so would help clarify how historical absence, procurement barriers, cost, cold-chain requirements, and limited clinician familiarity contribute to ongoing and future challenges in delivering complete PEP. A clearer, more focused discussion of RIG would strengthen the manuscript by providing a more realistic and policy-relevant assessment of barriers to comprehensive rabies prevention in Tanzania.

6. PLOS authors have the option to publish the peer review history of their article (what does this mean?). If published, this will include your full peer review and any attached files.

**Do you want your identity to be public for this peer review?** For information about this choice, including consent withdrawal, please see our Privacy Policy.

Reviewer #1: **Yes:** Prof. Dr Anurag agarwal

Reviewer #2: **Yes:** Sergei Vyatcheslavovitch Generalov

Reviewer #3: **Yes:** Aniruddha V. Belsare

Figure Resubmissions:

---

## [Decision Letter · Decision Letter 1]

2 Mar 2026

PGPH-D-25-03424R1

Practitioner’s perspectives on access to Rabies Post-Exposure Prophylaxis in Tanzania: A mixed-methods and theoretically-informed study to inform policy and practice

Dear Dr. Lushasi,

Thank you for submitting your manuscript to PLOS Global Public Health. After careful consideration, we feel that it has merit but does not fully meet PLOS Global Public Health’s publication criteria as it currently stands. Therefore, we invite you to submit a revised version of the manuscript that addresses the points raised during the review process.

**The manuscript has been evaluated by three reviewers, and their comments are available below.**

**The reviewers have a few remaining concerns that need attention. Specifically, the reviewer would like further discussion regarding the need to decentralize PEP services to priority facilities in the conclusions and tables.**

**Could you please revise the manuscript to carefully address the concerns raised?**

We look forward to receiving your revised manuscript.

Kind regards,

Katherine Demi Kokkinias, Ph.D.

Staff Editor

Journal Requirements:

Additional Editor Comments (if provided):

Reviewers' comments:

Reviewer's Responses to Questions

**Comments to the Author**

1. If the authors have adequately addressed your comments raised in a previous round of review and you feel that this manuscript is now acceptable for publication, you may indicate that here to bypass the “Comments to the Author” section, enter your conflict of interest statement in the “Confidential to Editor” section, and submit your "Accept" recommendation.

Reviewer #1: All comments have been addressed

Reviewer #2: All comments have been addressed

Reviewer #3: All comments have been addressed

2. Does this manuscript meet PLOS Global Public Health’s publication criteria? Is the manuscript technically sound, and do the data support the conclusions? The manuscript must describe methodologically and ethically rigorous research with conclusions that are appropriately drawn based on the data presented.

Reviewer #1: Yes

Reviewer #2: Yes

Reviewer #3: Yes

3. Has the statistical analysis been performed appropriately and rigorously?

Reviewer #1: Yes

Reviewer #2: Yes

Reviewer #3: N/A

4. Have the authors made all data underlying the findings in their manuscript fully available (please refer to the Data Availability Statement at the start of the manuscript PDF file)?

Reviewer #1: Yes

Reviewer #2: Yes

Reviewer #3: Yes

5. Is the manuscript presented in an intelligible fashion and written in standard English?

Reviewer #1: Yes

Reviewer #2: Yes

Reviewer #3: Yes

6. Review Comments to the Author

Reviewer #1: good article

Reviewer #2: (No Response)

Reviewer #3: Thank you for addressing reviewer comments from the previous round.

7. PLOS authors have the option to publish the peer review history of their article (what does this mean?). If published, this will include your full peer review and any attached files.

**Do you want your identity to be public for this peer review?** For information about this choice, including consent withdrawal, please see our Privacy Policy.

Reviewer #1: **Yes:** Prof. Dr Anurag Agarwal

Reviewer #2: No

Reviewer #3: **Yes:** Dr. Aniruddha V. Belsare

 Figure Resubmissions:

---

## [Decision Letter · Decision Letter 2]

10 Apr 2026

Practitioner’s perspectives on access to Rabies Post-Exposure Prophylaxis in Tanzania: A mixed-methods and theoretically-informed study to inform policy and practice

PGPH-D-25-03424R2

Dear Dr Lushasi,

We are pleased to inform you that your manuscript 'Practitioner’s perspectives on access to Rabies Post-Exposure Prophylaxis in Tanzania: A mixed-methods and theoretically-informed study to inform policy and practice' has been provisionally accepted for publication in PLOS Global Public Health.

Best regards,

Julia Robinson

Executive Editor

Reviewer Comments (if any, and for reference):

Reviewer's Responses to Questions

**Comments to the Author**

1. If the authors have adequately addressed your comments raised in a previous round of review and you feel that this manuscript is now acceptable for publication, you may indicate that here to bypass the “Comments to the Author” section, enter your conflict of interest statement in the “Confidential to Editor” section, and submit your "Accept" recommendation.

Reviewer #3: All comments have been addressed

2. Does this manuscript meet PLOS Global Public Health’s publication criteria? Is the manuscript technically sound, and do the data support the conclusions? The manuscript must describe methodologically and ethically rigorous research with conclusions that are appropriately drawn based on the data presented.

Reviewer #3: Yes

3. Has the statistical analysis been performed appropriately and rigorously?

Reviewer #3: Yes

4. Have the authors made all data underlying the findings in their manuscript fully available (please refer to the Data Availability Statement at the start of the manuscript PDF file)?

Reviewer #3: Yes

5. Is the manuscript presented in an intelligible fashion and written in standard English?

Reviewer #3: Yes

6. Review Comments to the Author

Reviewer #3: All comments have been addressed.

7. PLOS authors have the option to publish the peer review history of their article (what does this mean?). If published, this will include your full peer review and any attached files.

**Do you want your identity to be public for this peer review?** For information about this choice, including consent withdrawal, please see our Privacy Policy.

Reviewer #3: **Yes:** Aniruddha V. Belsare
